# Heterosexual Intimate Partner Femicide: A Narrative Review of Victim and Perpetrator Characteristics

**DOI:** 10.3390/brainsci15060589

**Published:** 2025-05-29

**Authors:** Anastasia Koureta, Manolis Gaganakis, Eleni Georgiadou, Vasilios P. Bozikas, Agorastos Agorastos

**Affiliations:** 1II. University Department of Psychiatry, School of Medicine, Faculty of Health Sciences, Aristotle University of Thessaloniki, 56430 Thessaloniki, Greece; 2School of Medicine, Faculty of Health Sciences, Aristotle University of Thessaloniki, 56430 Thessaloniki, Greece; 3Northwestern District Center for Mental Health, Psychiatric Hospital of Thessaloniki, 56430 Thessaloniki, Greece

**Keywords:** intimate partner violence, femicide, abuse, psychological profile, mental health

## Abstract

**Background:** Intimate partner femicide (IPF) is the most common form of femicide and a severe expression of gender-based violence, highlighting persistent gender inequality worldwide. Addressing this major public health concern requires a comprehensive synthesis of existing evidence to inform prevention strategies. This review aims to identify risk factors for IPF and explore the demographic, behavioral, and psychosocial characteristics of victims and perpetrators. **Methods:** A narrative review was conducted using a systematic literature search in Scopus, Web of Science, and PubMed, following PRISMA guidelines. Studies were selected based on predefined inclusion and exclusion criteria. Out of 1200 identified records, 51 met the criteria and were included. Data extraction and analysis were conducted independently by two reviewers. Findings are presented narratively. **Results:** The review identified multiple risk factors for IPF. Victims were more likely to be married, with a history of psychological and physical abuse as well as substance use. Perpetrators were typically older, with higher rates of unemployment, psychiatric disorders, and substance use. Common precipitating factors included jealousy, separation, and recurrent conflicts. Weapon use—particularly knives and firearms—and “overkill” were frequent. Perpetrators often exhibited stalking behaviors and a history of intimate partner violence. Compared to other homicide offenders, IPF perpetrators were generally older, more often employed, and less likely to have a criminal background, but more likely to engage in intimate partner violence and hold patriarchal beliefs. **Conclusions:** IPF is not an unpredictable act. Despite the heterogeneity among perpetrators, identifiable risk indicators can inform effective prevention and intervention efforts.

## 1. Introduction

Intimate Partner Violence (IPV) represents a pervasive global issue, encompassing behaviors by a current or former partner that cause physical, sexual, or psychological harm [1]. This includes acts of physical aggression, emotional abuse, and coercive control, which collectively undermine the safety and well-being of victims [1]. IPV affects individuals across diverse social backgrounds, cultural groups, sexual orientations, and gender identities, with significant implications for public health and society [2]. While both men and women can be victims, women in heterosexual relationships disproportionately experience severe consequences [3]. Globally, approximately one in four women has experienced IPV, underscoring its widespread nature and profound impact [4].

IPV can escalate from emotional and physical violence to homicide, i.e., the murder of one partner by a current or former intimate partner. Notably, a significant proportion (up to 55%) of all homicides globally are committed by intimate partners [2]. Thereby, women are the predominant victims, and approximately 60% of female homicides are committed by a male intimate partner [2]. In contrast, corresponding figures for male victims are substantially lower in European contexts, emphasizing gender-specific patterns and disparities in victimization and perpetration rates [5].

Especially the murder of the female partner (femicide) by a male current or former intimate partner is a critical aspect of IPV, reflecting the ultimate consequence of gender-based violence. Although historically treated as a private matter across many cultures, heterosexual intimate partner femicide (hIPF) has garnered increasing attention as a severe crime and social problem demanding systematic investigation and intervention [5]. Despite overall declining trends in homicide rates, hIPF rates remain alarmingly stable, highlighting persistent challenges in prevention and protection efforts [1].

Numerous theoretical perspectives have been advanced to elucidate the complex dynamics of IPV and hIPF. While general violence theories highlight individual-level risk factors and criminal behavior, feminist theories emphasize the role of patriarchal ideologies and gender inequalities in perpetuating violence against women [6]. These perspectives underscore how societal norms and power dynamics contribute to the manifestation and perpetuation of violence within intimate relationships. Additionally, social disorganization theories offer insights into community-level factors influencing violence, suggesting that neighborhood characteristics such as high crime rates and social fragmentation may contribute to increased risk [7]. On the other hand, evolutionary psychology theories (e.g., male sexual proprietariness theory) propose that jealousy and threats to perceived reproductive control can escalate into fatal violence within intimate partnerships, particularly during periods of relationship strain [8].

The consequences of hIPF extend beyond individual victims to encompass broader social implications, as families and communities grapple with the enduring grief and societal costs associated with these tragic deaths [1]. Moreover, the pervasive nature of gender-based violence perpetuates cycles of fear and insecurity, undermining efforts to achieve gender equality and social justice.

This review aims to systematically explore factors associated with hIPF, focusing specifically on murders committed by male intimate partners and female victims within heterosexual relationships [1]. The primary objectives of this study are to (i) identify societal, demographic, and relationship-related factors associated with both perpetrators and victims of IPF, (ii) examine the psychological characteristics and mental health issues of both perpetrators and victims, and (iii) explore behavioral and psychological typologies of IPF offenders based on patterns identified in the existing literature, highlighting differences from other types of homicide offenders. By addressing these objectives, this review seeks to contribute to a deeper understanding of the multifaceted nature of intimate partner femicide, informing prevention strategies, policy development, and interventions aimed at reducing violence and protecting vulnerable individuals.

## 2. Materials and Methods

This narrative review employed a systematic literature search strategy to synthesize findings from heterosexual intimate partner femicide (hIPF) studies, aiming to identify patterns across diverse sources [9]. The review process was informed by PRISMA guidelines, although certain elements (e.g., protocol registration, risk of bias assessment) were not applied, given the narrative scope.

A structured search was conducted across three electronic databases—Scopus, Web of Science, and PubMed—for studies published between 1990 and 2023. Search terms included combinations of “femicide/homicide”, “intimate partner”, and “risk/factors/characteristics/mental health” and similar terms, targeting titles and abstracts. Reference lists of relevant articles were also hand-searched to capture additional studies. The last search was performed in December 2023. Two independent reviewers screened titles and abstracts for eligibility, with discrepancies resolved through discussion. The selection process is depicted in a PRISMA flow diagram (Figure 1).

Inclusion criteria were as follows:(A)Studies focusing on hIPF cases perpetrated by males in heterosexual relationships, including both completed femicides and serious attempts, based on evidence of shared characteristics between the two [10,11]. Studies with mixed samples (e.g., female perpetrators or same-sex couples) were excluded unless they separately reported findings on male-perpetrated hIPF [12].(B)Observational studies examining risk factors, as well as perpetrator and victim characteristics. Case reports, reviews, and studies focusing exclusively on the aftermath or impact of femicide were excluded.

This approach allowed for the identification of recurring patterns across psychological, demographic, and criminological domains in both perpetrators and victims, as well as murder typologies and relevant social contexts.

## 3. Results

### 3.1. Search Results

The initial search identified 1200 studies on partner homicide, 445 risk factor-related studies, and 209 mental health-related studies. After applying selection and exclusion criteria, 51 studies were included, encompassing a research project from Australia [13] (Table 1). The study by Chimbos et al. [14] was excluded due to a mixed population sample lacking separate results. The review included data from twelve studies conducted in the USA, seven in the UK, five in Spain, five in Canada, five in Sweden, three in Portugal, three in Australia, two in Finland, two in Turkey, and one study each from Norway, Denmark, Ghana and Italy. Studies from other regions (e.g., South Africa, India, China) did not meet the inclusion criteria. The review spans studies from 1998 [15] to 2024 [16]. Table 1 offers a comprehensive overview of all 51 included studies, detailing authors, years, countries, study samples, methods, data sources, descriptive analyses, main factors, and mental health results.

### 3.2. Study Characteristics

Most studies involved descriptive analysis, with some focusing on risk factors without comparing other homicide categories [8,19,21,24,25,27,28,29,30,31,33,36,39,43,47,50,52,56]. Other studies compared subgroups such as attempted hIPF [10,11] and perpetrator suicide [21]. Some studies compared male with female perpetrators of intimate partner femicide [5,22,32,35,45] or with male perpetrators of other femicides [15,18,34,38,42]. Others investigated generally violent behavior in hIPF [4,18,23,44,47,55]. Some studies compared hIPF perpetrators/victims with those of physical abuse by partners [10,11,16,20,40,51,53,57,58]. Fewer studies used control groups from the general population [11,41,51].

### 3.3. Data Sources

Official databases were used in some studies, such as the National Violent Death Reporting System [29,30,32], the Wisconsin Violent Death and Domestic Violence Archive [42], the National Homicide Database and National Hospital Record in Sweden [4,23,41], the Finnish Homicide Monitoring System [44], and the national homicide program in Australia [18]. Other data sources included special committees (Ontario Domestic Violence Death Review Committee, Canada; ANROWS, Australia) [13] and detailed court, police, forensic, medical, and psychiatric records [10,13,18,42,47,49,52,55]. Five studies used data from newspapers and media [8,21,26,43,46]. Psychiatric assessments were included in some studies [4,25,28,35,45,52,55,58]. Toxicology tests were performed in some cases [4,11,15,23,28,29,39]. Interviews with perpetrators were conducted in several studies [4,10,11,16,18,19,20,25,27,35,40,47,49,50,52,53,55,57,58]. Only Pineda et al. [16] included interviews with offenders’ relatives. Some studies included interviews with survivors of attempted homicide [56] or victims’ relatives [11,16,19,47,49,51,57]. Finally, structured questionnaires for assessing risk of partner violence including Danger Assessment [35,56,57], Spousal Assault Risk Assessment (SARA) [23,40], SIVPAS [35], RisCanvi [34], VPER [19], while questionnaires for assessing mental health issues including MCMI-II [58], PEN model of Personality Eysenck [16,19], Toronto Alexithymia Scale (TAS-20) [20] and Hare Psychopathy Checklist Revised (PCL-R) [10,45,53,55] were also utilized in various studies.

### 3.4. Perpetrator-Related Findings

#### 3.4.1. Socio-Demographic Characteristics

The average age of hIPF perpetrators ranges from 34 to 52 years across various studies (in most studies, 40–50 years, with a maximum age of 97). Offenders were older compared to male perpetrators of other homicides [15,18,23,34,38,45,49] and compared to perpetrators of IPV [40,53,57]. In most studies, perpetrators were older than victims in most couples [11,13,21,39,43]. The hIPF risk increases with increasing age difference between partners, specifically when the man is more than 16 years older than the victim, or the woman is 10 years older than the man [54]. IPF perpetrators, although often employed [4,10,43], exhibit higher unemployment rates or live more often on benefits compared to the general population [5,11,12,13,21,28,39,44,45,52,57] and compared to perpetrators of IPV [11,57]. However, they are more likely to be employed than offenders of other types of homicide [18,23,38,44,45,49] and show similar employment rates to male murderers who killed a relative [44]. Unemployment, according to the study by Campbell et al. [57] and David and Jaffe [24], is a risk factor for hIPF, while higher risk is also seen in perpetrators with professions related to law enforcement (9.7% of offenders) [21]. The perpetrators’ educational level is lower than that of the general population [41], and in most studies, it is below secondary education level [5,10,11,13,23,34,40,49,53,57]. On the other side, according to Dobash et al. [49], the educational level of hIPF offenders is higher compared to that of male murderers of other crimes but lower compared to perpetrators of IPV [57]. In addition, hIPF offenders generally show a socio-economic status that is below average [10,11,19,34,39,40,41,43,44,53,57], but is higher compared to male murderers of other crimes [49] and IPV offenders [10].

The prevalence of various racial and ethnic minorities has been discussed in the field of IPV, but only a few studies have investigated this issue in a pure population of heterosexual couples with male offenders and female victims. Sebire et al. [39] report high percentages (25%) of African-American offenders, which is significantly higher than the percentage in the general population of the study area, contrary to the reported results by Dobash et al. [49] from the same region (5%). In U.S. studies, high percentages of offenders were identified as African-American (489%) [11,57]. The Australian study by Boxall et al. [13] also recorded high percentages from indigenous population (Aborigines). In studies primarily from Scandinavia, offenders were more often migrants compared to the general population group [41,55], or there was increased representation of migrants in the hIPF perpetrator sample, although clear comparisons with the general population were not provided [13,23,28,44]. A study from Israel showed increased representation of Ethiopian migrants in the sample of hIPF offenders [36]. David and Jaffe [24] reported that migrants who had been in Canada for a longer period, especially those with pre-migration trauma history, showed a higher risk to conduct hIPF and also had a higher percentage of mental health issues. Finally, some studies indicate that hIPF offenders are more often migrants compared to other male homicide offenders [18].

#### 3.4.2. Personal History

An unstable family environment or/and being a witness or victim of domestic violence during childhood is often found in the personal history of hIPF offenders, with percentages ranging from 11.1% to 32% [13,45,49], but less frequently compared to IPV perpetrators of abuse [40] and other male homicide offenders [49]. Similarly, behavioral problems and delinquency before the age of 16 are less frequent compared to other homicide offenders [34,49]. hIPF perpetrators very often report relationship instability with many partners and a history of violent behavior in previous relationships [47,49]. In hIPF perpetrators, a criminal history or previous convictions is found in higher percentages compared to the general population, ranging from 16.3% to 72.6% [5,8,12,15,17,21,23,28,31,33,34,40,41,44,45,49,53,54,55,57]. A criminal history is an important risk factor for hIPF according to several studies [41,54]. However, a criminal history is less frequent in hIPF offenders compared to other male homicide offenders [18,23,44,45,49,55], or to other femicide offenders [15,34] or IPV male offenders [40], while Campbell et al. [57] report higher percentages compared to IPV offenders.

#### 3.4.3. Substance Use History

A history of substance use (excluding alcohol) is often found in hIPF offenders in percentages ranging from 10–65.4%, higher than in the general population [10,12,19,34,45,57]. Only the Turkish study reports a low percentage (5.1%) [38]. Alcohol abuse is found at even higher percentages (23.8–62.3%) in hIPF perpetrators [10,11,12,19,28,34,45,49,57], except for the study from Turkey (3.8%) [38]. Some studies provide data on all psychoactive substances, which again show elevated percentages compared to the general population, but there is great variation between studies, with percentages ranging from 5% to 64% of hIPF offenders [5,13,23,28,40,44,55]. Alcohol and substance use is also strongly linked to IPV and is considered an important risk factor for hIPF [57].

#### 3.4.4. Mental Health Issues

hIPF perpetrators’ positive mental health history, including psychiatric AXIS-I disorders and personality disorders, is considered a significant risk factor for intimate partner homicide [28,41]. Mental health history and suicidal ideation as risk factors for hIPF are also mentioned in the studies by Belfrage and Rying [55] from Sweden and Santos-Hermoso et al. [19] from Spain. A history of psychiatric hospitalizations is reported in 17% [41]–32% [23,45] of cases, while history of psychiatric diagnoses (including personality disorders) is reported in 3% [8]–60% [28,55] of hIPF perpetrators; in studies excluding personality disorders, the percentages range from 10% [38] to 19% [5]. Studies including national hospitalization records and psychiatric assessments show much higher percentages, such as 30.4% [4,23], 32.5% [12], 43% [13], 40–47% [17], 60% [28,55], and even around 80% [45]. Exceptions to this trend are reported by Velopulos et al. [32] and Lysell et al. [41], whose findings are based exclusively on data from psychiatric hospitalizations. In their studies, only 7.5% and 12.6% of hIPF perpetrators, respectively, had a documented history of psychiatric disorders.

With respect to the diagnosis history of hIPF perpetrators, most studies report percentages below 10% for psychosis history [4,13,33,41,45], while other studies report percentages of 13% [28] and 36% [55]. On the other hand, history of mood disorders, mainly depression, are reported to range from 1% to 64% [4,13,23,25,28,41,45,55]. Studies reporting results from psychiatric-judicial evaluations report the prevalence of personality disorders, ranging from 3% to 70% [4,13,33,34,41,45,55], while some studies report that up to 10% of hIPF perpetrators were evaluated as not criminally responsible on account of serious mental disorder [35,44,45]. Some studies report similarities in mental health history with IPV offenders [10,40], female intimate partner murderers [35], and other male homicide offenders [49].

Suicide or serious suicide attempts by hIPF perpetrators are particularly frequent after the crime, but the rates vary across studies, ranging from 5% to 50% [4,5,12,16,17,21,25,30,31,35,36,41,55,58]. In the study by Santos-Hermoso [19], suicide attempts range from 9.1% in the violent group to 35.5% in the high psychopathology group. Previous suicide attempts, suicidal ideation, or self-harm are recorded in 5–28.2% of hIPF perpetrators [12,34,41,45,57] while according to Rye and Angel, this as high as 52% [28]. Intimate partner murderers, compared other male homicide offenders, are more likely to commit suicide [4,55] and show more frequent suicidal or homicidal ideation compared to IPV offenders [40].

#### 3.4.5. Psychological Characteristics

Pineda et al. [16] found that hIPF perpetrators, compared to IPV offenders, exhibit higher levels of neuroticism (increased anxiety, worry, difficulty managing emotions) and psychoticism (lack of empathy, antisocial behavior, aggressiveness) and lower levels of extraversion. These results were also confirmed by Santos-Hermoso et al. [19]. Several studies also investigated psychopathy in hIPF perpetrators, showing low psychopathy scores [10,53,55]. Only Weizmann-Henelius [45] reported higher levels (23.5% showed a score above 25 on PCL-R), though this was still lower compared to the scores of other male homicide offenders. In comparison to IPV offenders, no significant differences in psychopathy were reported [10]. Investigating personality traits in hIPF perpetrators, Loinaz et al. [34] described impulsivity and emotional instability in 57%, hostility in 42.9%, irresponsibility in 57%, while Dobash and Dobash reported problems in romantic relationships with women in 75% of the population studied [47].

hIPF perpetrators also show differences in history of personality disorder diagnosis between IPV and other male homicide offenders. For example, Loinaz et al. [34] found that, compared to IPV offenders, hIPF perpetrators more frequently show personality disorders with passive-aggressive, dependent, self-harming, and avoidant traits, while IPV offenders tend to have antisocial and sadistic traits [58]. According to Vignola-Lévesque and Léveillé [20], hIPF perpetrators are more often nonalexithymic compared to IPV offenders, meaning they do show some difficulty with emotional awareness and expression—but to a lesser extent than IPV offenders, who are more frequently classified as alexithymic (i.e., with marked deficits in identifying and verbalizing emotions and engaging in outward, functional thinking).

#### 3.4.6. Different General Subtypes

Studies examining hIPF perpetrator characteristics in terms of criminal history, prior violent behavior, mental health history, and current psychopathology (i.e., presence of mental disorders and suicidal ideations), report similar offender general subtypes [19,52]. Specifically, 15.3% [52] to 23.4% [19] of the total cases involved hIPF perpetrators with no psychopathological traits or history of violent behavior. However, most offenders in both studies had a history of violent behavior/criminal activity, with or without moderate to high psychopathology. On the contrary, the study by Santos-Hermoso [19] showed a significant percentage of hIPF perpetrators without criminal/violent history but with high psychopathology. In the study of Vignola-Lévesque and Léveillée [20], two “subtypes” of offenders are described: one with dominant psychopathological traits (history of suicide attempts, indicating psychological distress), without criminal/violent history, who recently experienced a breakup, and another with violent and controlling traits, criminal/violent history and lower likelihood of previous suicide attempts. A violent and partner-controlling subtype of hIPF perpetrators has also been identified in other studies [50,58]. In the study by Boxall [13], three subtypes of offenders were identified: obsessive, with jealous and controlling behaviors, threatened by separation; offending, often from a minority group, with mental issues and a history of criminal activity; and older, with both physical and mental health problems.

#### 3.4.7. Personal Beliefs

Regarding personal beliefs, Eriksson et al. [18] compared hIPF perpetrators with other femicides and other male homicide offenders and reported that both groups of femicides, in contrast to other male homicide offenders, believe they have more rights than their partner, justify abuse, show possessiveness, and need for control over their partner (behaviorally, socially, or over information). Similar results were found for hIPF perpetrators and IPV offenders by Férnandez-Montalvo et al. [53]. Finally, Dobash and Dobash [47] reported distorted beliefs about male dominance in intimate relationships, female submission, and the need for control, punishment, and correction of behavior if the woman does not comply with the man’s demands in hIPF perpetrators. Finally, Dobash [47] and Cunha and Gonçales [40] found that a large percentage of perpetrators do not express remorse (36%), do not show empathy towards the victim (49%) and deny responsibility for the hIPF, holding the victim or external factors accountable.

#### 3.4.8. Comparison of hIPF Perpetrators with Other Male Homicide Offenders

Only two Scandinavian studies [41,44] included general population controls. These studies found that hIPF perpetrators had lower socio-economic and educational levels, were more often unemployed, had a history of psychiatric hospitalizations, self-destructive behavior, substance use, and prior convictions. Compared to other male murderers, intimate partner murderers were generally older, more often employed, migrant, and less likely to have substance use disorders, criminal histories, or psychopathy. However, they had a greater history of intimate partner violence and were more likely to commit suicide post-crime [4,18,44,45,49,55]. They also held more patriarchal beliefs [18] and had fewer adverse childhood experiences [49]. Intimate partner murders were more often committed at home and by strangulation.

When compared to other femicide offenders, intimate partner murderers had lower rates of criminal activity, substance use, psychopathy, and childhood adversity, but were older and more likely to be employed. Victims were more often married and of Black ethnicity [15,18,34,38,42,45]. Other femicides occurred more frequently in urban, disadvantaged areas [15,42]. Studies comparing murderers with abusive partners found that intimate partner murderers were older, African American, and less likely to have a criminal history but more often possessed a weapon or had threatened with one [10,11,40,57]. Psychological findings were inconsistent, though some studies reported higher neuroticism, psychosis, and lower extraversion in femicide offenders [16,20,58], while others found no significant differences [53]. Comparisons between male and female intimate partner murderers suggest that female perpetrators more often acted in self-defense, had victimized partners with substance abuse histories, and were less likely to die by suicide [5,35,45]. Some studies found higher rates of mental illness and substance use in female perpetrators, though results were mixed [22,32,35].

### 3.5. Victim-Related Findings

Fewer studies have focused on hIPF victims, and most data originate from relatives of the victims or from severe attempted hIPF survivors.

#### 3.5.1. Socio-Demographic Characteristics

hIPF victims have been reported to be more often married compared to other victims of femicides [15,38]. In studies from the USA and UK, Black women show an increased risk of being murdered by their intimate partner [39,42,46,54,56,57]. In other studies, a similarly higher risk is reported for Hispanic [46] or indigenous (Aboriginal) women [8]. Among hIPF victims, higher rates of migrant women compared to the general population have been reported [28,42,46,55]. The majority of hIPF victims had a secondary education level or lower [42,57]. According to Sharps et al. [11], hIPF victims often were in a better employment situation than the perpetrators, while in contrast, Gillespie and Reckdenwald [37] reported financial difficulties of women as a factor associated with an increased hIPF risk.

#### 3.5.2. Personal History

Women have a reduced risk of being an hIPF victim during pregnancy and the first year after birth [51]. However, in cases of abuse history during pregnancy, the risk of being killed by the intimate partner increases [57]. hIPF victims have a criminal history in 19.1% of cases [12].

#### 3.5.3. Substance Use

hIPF victims are often found to be under the influence of alcohol (in 11–66% of cases) [11,32,35,39,42,45,49,55,57], and were less frequently under the influence of other substances [35,39]. hIPF victims had a history of substance use or alcohol in 10.5–50% of cases [11,23,35,57].

#### 3.5.4. Mental Health Issues

History of mental illness is reported in the study by Rye and Angel [28] in 14% of hIPF victims, with the most common diagnosis being depression, while lower rates are reported in other studies [21,38,39]. In the study by Pineda et al. [16], personality assessment through interviews from relatives of the hIPF victims revealed that the victims had low levels of neuroticism and psychoticism, and high levels of extraversion, with no differences compared to victims of IPV. In a qualitative study by Nikolaidis et al. [56], 50% of the women who survived a severe hIPF attempt did not perceive the danger.

Table 2 summarizes the key findings from the reviewed studies, categorizing them according to the various factors related to both victims and perpetrators of hIPV, providing a concise overview of the patterns observed across the literature.

### 3.6. Relationship-Related Factors

Firstly, in the majority of hIPF cases, the victim was the current partner or spouse [10,13,19,21,24,26,28,29,31,40,43], and the couple cohabited [12,13,28,57]. In many cases, there is a history of violent behavior towards the partner, with percentages ranging from 36 to 91% [4,5,8,12,13,17,18,20,21,28,32,39,40,44,45,49,55,56,57,58]. Some studies mention a high percentage of previous official IPV reports from the victim, prior IPV-related convictions of the partner, or even previously issued protective orders [12,55,57]. However, Rye and Angel [28] mention that only 29% of hIPF victims had previously filed police reports, while McLachlan [17] reports low rates of seeking help from domestic violence services. Several studies also show that an IPV history is one of the most significant risk factors for hIPF [19,31]. IPV is more common among hIPF perpetrators compared to other male homicide offenders [18,49]. Other studies report that previous life threats (towards the victim or the victim’s family) and homicidal ideation often precede hIPF [12,15,19,43,55,57], reaching up to 83.4% of cases, while other studies report previous homicidal ideation in only 31% of cases [28]. In addition, history of psychological abuse towards the victim, forced sexual acts and strangulation attempts by the male partner have been reported in 82.7% [35], 57.1%, and 56.4% of the couples [57]. In their qualitative study, Dobash and Dobash [47] reported that the relationship was often characterized by possessiveness, conflicts, controlling behavior, and abuse, findings later confirmed by McLachlan [17] and Nicolaidis et al. [56]. In general, frequent arguments and conflicts have been repeatedly reported to precede hIPF [21,33,36,43,49], while there are also some reports on stalking behavior preceding femicide [12,15], which is also considered a risk factor for hIPF [19,57]. Interestingly, only a few studies investigate the history of IPV by the female hIPF victim in the relationship; those studies offer various reports, from 6% to around 20% of couples [12,44,49], while Vatnar et al. [35] report mutual violence in 52% of cases.

Regarding the duration of the relationship, most studies suggest that most couples had long-term relationships, with 80% of couples being together for more than a year [28,57], and 62% for more than three years [12], while other studies [13,35] report an average relationship duration of 10 years. In addition, presence of children from the hIPF perpetrator [19,21,57] or from another father [8,39] in the household is very often reported, while in other studies, presence of children from another father in the household is considered a risk factor for hIPF [57]. Similarly, recent separation or threat of separation by the female partner is identified in many cases as a triggering factor [5,10,13,17,19,21,23,27,28,31,33,36,38,43,49,52,55,56,57,58] and a very important overall risk factor for hIPF [10,57]. However, Nicolaidis et al. [56] emphasize that separation is mostly not a result of IPV by the male partner violence of the perpetrator, but rather of other reasons (e.g., financial, substance use). Finally, jealousy and/or infidelity [8,21,23,28,31,32,33,36,38,43,45,49,55,57], along with arguments, are frequently identified as triggers for hIPF.

### 3.7. Circumstantial Factors

In studies from most European countries [19,23,28,31,34,35,36,45,49,55] and Australia [13,17], knives or other sharp utensils are the most commonly used weapons, whereas firearms are more prevalent in studies from the USA, Ghana, Turkey, and one study from Italy [8,15,21,29,30,32,38,42,43,46,57]. Weapon possession itself has been identified as a significant hIPF risk factor [10,57], with hIPF perpetrators more frequently owning weapons compared to perpetrators of non-lethal intimate partner violence [40,57].

Murder by multiple blows or gunshots—often described as “overkill”—is frequently reported in hIPF cases, indicating a high emotional charge or “crime of passion” [8,29,31,36,38,42,49,58]. Strangulation is also a common method, although its prevalence varies across studies [4,19,23,28,31,34,46]. The crime scene was most often the couple’s shared home or, less commonly, the victim’s parental home [34,43].

Premeditation appears to be rare: only a few studies report planned killings [5,43], whereas the majority suggest that hIPFs are more often impulsive acts without prior planning [17,23,28,31,58].

### 3.8. Community-Related Factors

Studies from the USA report a higher frequency of hIPF in rural compared to urban (metropolitan and non-metropolitan) areas [17,37] and compared to other types of femicides [15]. This is probably related to the higher proportion of women in financial distress and the insufficient services available for abused women (e.g., shelters, helplines, etc.). On the contrary, an Australian study by McLachlan [17] found no significant differences in hIPF numbers between rural and urban areas, nor in access to support structures. Reckdenwald et al. [29] reported that, in rural areas, hIPFs were more often committed with a firearm, and the hIPF victim was more likely to be white and of older age. Interestingly, Sivaraman et al. [30] found that the frequency of hIPF in the USA was lower in states with more restrictive gun ownership laws. Several studies have examined the impact of the COVID-19 pandemic on femicide rates, yielding mixed findings. Aebi et al. [59] and Cantor et al. [60] reported no increase—or even a decline—in femicide rates during the early pandemic period in several Spanish-speaking countries and in Chile, respectively. Similarly, Asik and Nas Ozen [26] found that in Turkey, hIPF cases declined by 57% during pandemic-related restrictions and by 83.8% during full lockdowns, with the sharpest decrease observed in killings by former partners. However, country-specific trends highlight notable variation. In Greece, Karakasi et al. [61] reported that while the overall number of homicides declined in 2021, domestic homicides rose to the highest annual figure ever recorded nationally, with a disproportionate increase in female victims. The authors suggest that this surge may be linked to increased media attention on femicide, the effects of the financial crisis, and rises in alcohol and drug use during the pandemic.

## 4. Discussion

This review systematically synthesizes existing literature on heterosexual Intimate Partner Femicide (hIPF), highlighting multifactorial risk dynamics. Key perpetrator-related risk factors include unemployment, prior criminality, substance use, psychiatric disorders, and suicidal ideation [5,62,63]. These elements indicate that hIPF often extends beyond the realm of interpersonal conflict, reflecting broader issues of psychosocial instability and maladaptive coping mechanisms.

While certain victim-related characteristics—such as immigrant status, minority ethnicity, or socio-economic disparities—are mentioned in the literature [63], they do not consistently reach statistical significance in multivariate models. This underlines a key insight: although certain profiles may be overrepresented, hIPF can affect women across socio-demographic strata.

Methodological limitations, particularly the inability to interview deceased victims, constrain the depth of understanding regarding victim characteristics. Nevertheless, qualitative methodologies such as interviews with family members or acquaintances, and psychological autopsy techniques—structured approaches that retrospectively reconstruct the psychological state of the deceased using multiple data sources (e.g., medical records, police reports, and informant interviews)—offer promising avenues for advancing victim profiling [64]. These methods can provide a more nuanced understanding of the psychosocial dynamics and warning signs preceding hIPF, thereby contributing to more informed prevention strategies.

Regarding the relational context, the presence of controlling behaviors, threats, previous violence, and ideation of homicide or suicide appear to be critical indicators of elevated risk. These findings support a conceptualization of hIPF not as an unanticipated or isolated act, but as the culmination of coercive and escalating abuse over time. While most homicides are not premeditated in a legal or tactical sense, they rarely occur without prior warning signs, especially in the context of acute stressors such as separation, jealousy, or infidelity [5,62,63].

At the community and societal levels, increased rates in rural areas have been attributed to factors such as firearm availability and geographic isolation. The impact of gender equality indicators remains mixed; some evidence suggests that perceived threats to traditional gender hierarchies may fuel violent behavior among certain male perpetrators.

Although the literature presents various typologies of hIPF perpetrators, these are often inconsistent or overlapping [13,19,20,50,52,65]. Based on the findings of this review, we propose a conceptual grouping into three broad profiles that may help illustrate common patterns. The first includes individuals with a history of violent behavior, often involving prior intimate partner violence or criminal convictions, typically accompanied by moderate psychological disturbance. The second consists of perpetrators with severe mental illness, such as psychosis, mood disorders, or complex personality pathology, often without a history of criminal behavior. The third group comprises socially conventional men whose violence is primarily driven by controlling, jealous, or patriarchal attitudes rather than diagnosable psychopathology. Although these profiles are not mutually exclusive and may overlap, they provide a useful framework for understanding perpetrator variability and tailoring prevention and intervention strategies accordingly.

The role of distorted gender ideologies is underscored across studies, with many perpetrators exhibiting beliefs supportive of male dominance and female subordination. These ideological underpinnings often manifest in patterns of surveillance, restriction, and control prior to the homicide.

General Strain Theory [66] offers a useful interpretive lens, emphasizing how structural stressors—such as financial hardship, relationship dissolution, or perceived betrayal—interact with individual vulnerabilities (e.g., mental illness, emotional dysregulation) to precipitate violent outcomes.

Finally, the review reinforces that intimate partner violence often precedes femicide, underscoring the importance of early detection. Proactive risk assessment, the use of validated tools (e.g., Danger Assessment, B-SAFER) [67,68] and coordinated intervention strategies are essential. Targeted responses—especially in high-risk groups such as women in recent separation or with pregnant status, and men with weapon access or psychiatric comorbidities—are crucial to effective prevention. Collaboration between law enforcement, health, and social services remains a cornerstone in reducing the incidence of hIPF.

Future research should explore femicide in diverse couple types and improve gender-based violence records globally. Studies comparing femicide in heterosexual couples with same-sex couples or involving individuals with diverse gender identities, such as transgender or non-binary individuals, could provide a more comprehensive understanding of the phenomenon. Additionally, improving data accuracy and consistency in records on gender-based violence across different countries will offer a clearer picture of the problem and aid in developing more effective prevention strategies.

## 5. Conclusions

Understanding intimate partner femicide requires a multifaceted approach that accounts for psychological, relational, and socio-cultural dynamics. While methodological limitations persist, especially regarding victim-related data, advancing research tools and cross-sector collaboration can enhance insight into risk factors. Crucially, the findings highlight the importance of proactive prevention strategies. These include timely risk assessment, early intervention in cases of intimate partner violence, and coordinated responses among law enforcement, social services, and mental health professionals. Strengthening these measures is essential for identifying women at risk and ultimately preventing future femicides.

## Figures and Tables

**Figure 1 brainsci-15-00589-f001:**
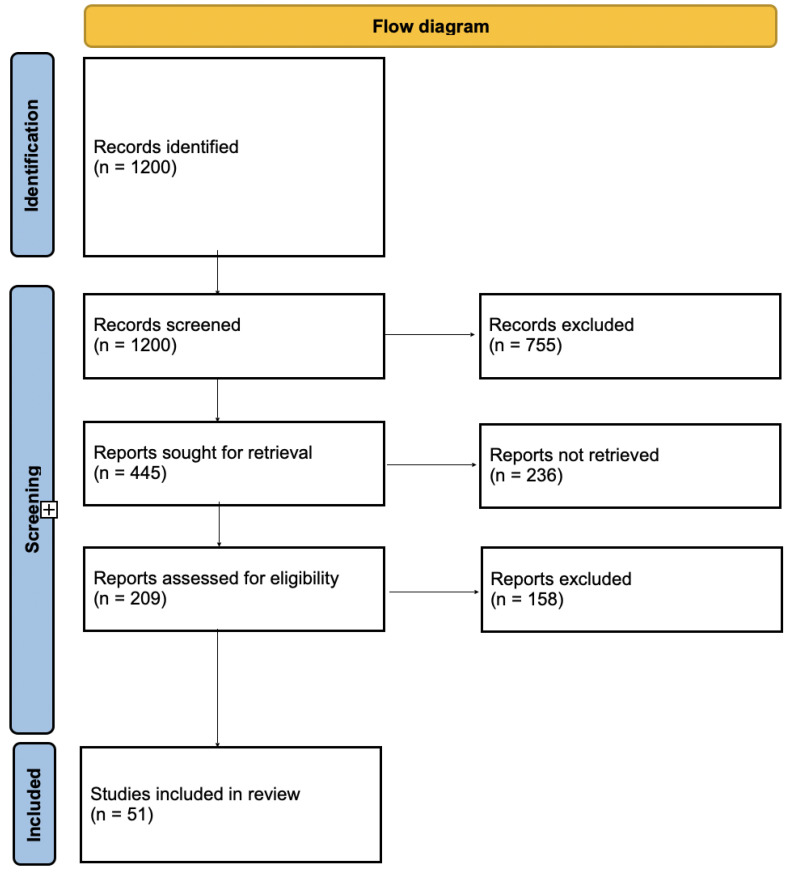
Flow chart diagram of study selection procedure for inclusion.

**Table 1 brainsci-15-00589-t001:** Basic characteristics of included studies.

Author (Country)	Sample	Data/Methodology	Descriptive Analysis	General Findings	Mental Health-Related Findings
[16] (Spain)	169 hIPF perpetrators, 165 hIPF victims, 110 IPV perpetrators, 106 IPV victims	Semi-structured interviews with perpetrators, victims, and relatives of deceased perpetrators and victims. Psychological autopsy method, Eysenck’s Pen model of Personality questionnaire. Comparison of the four groups for personality traits	Perpetrators: avg age 46 years, 71.9% Spanish. Victims: avg age 42 years, 68.4% Spanish.	Perpetrators had higher neuroticism, psychoticism, lower extroversion. 25% committed suicide.	Higher levels of neuroticism and psychoticism, lower extroversion in perpetrators.
[17] (Australia)	100 hIPF	Records from a special domestic violence service. Retrospective study comparing rural and urban homicides	Increased ratio of IPF in rural areas	Similar characteristics in rural and urban areas; separation more common trigger in urban. In total, 13.2% rural, 17% urban perpetrators attempted suicide.	High substance use (45–49%), history of mental disorder (40–47%).
[18] (Australia)	68 hIPF, 44 other femicides, 135 male perpetrator and victim	National Homicide Monitoring Program data. Interviews with perpetrators, questionnaires on childhood experiences, substance use history, previous violence, beliefs about marriage roles, relationship rights, partner violence, and jealousy. Comparison of the three groups on all variables.	-	IPF group older, non-British, more partner violence history, less criminal history.	Less substance use compared to men killers. More possessiveness, need to control.
[19] (Spain)	171 hIPF	Police, judicial records, interviews with perpetrators and relatives of victims, psychological autopsy, risk assessment questionnaires (VPR, VPER), Eysenck’s personality questionnaire. Electronic template with 105 variables.	Victim avg age 41.9 years, 68.4% Spanish. Perpetrator avg age 46.2 years, 71.9% Spanish, low socio-economic.	Significant risk factors include previous violence, work problems, relationship break.	Four perpetrator types: normal, high psychopathology, violent, violent and high psychopathology.
[20] (Canada)	22 hIPF, 45 IPV	Interviews with perpetrators and judicial records. TAS-20 Alexithymia Questionnaire.	Avg age 52.2 years, history of partner violence (63.6%).	Two perpetrator types: self-destructive, no criminal history, recent separation; violent, controlling, criminal history.	Most perpetrators sub-alexithymic.
[21] (Italy)	409 hIPF	Retrospective study of femicides in Italy from 2010–2019. Media databases. Descriptive analysis of factors in each group. Comparison of the two groups for triggers. Investigation with hierarchical logistic regression analysis of predictors for suicide.	Perpetrator avg age 51.5 years, 81.4% Italian, 35.2% unemployed.	Most common motive: refusal to accept separation. 82.4% no criminal history.	35.7% suicide in perpetrators. 12.6% substance/alcohol users.
[13] (Australia)	199 hIPF	Judicial, police, forensic records, and newspaper archives. Data on Australian perpetrators from 2007–2018. Excludes perpetrators who committed suicide and those deemed not responsible. Descriptive analysis and qualitative study.	Perpetrator avg age 41 years, 37% employed, 26% indigenous.	Issues: chronic mental health problems, traumatic experiences, recent separation.	Mental health issues in 43%. Depression 26%, psychosis 6%, personality disorder 7%.
[4] (Sweden)	46 hIPF, 133 male perpetrator and victim (MMH)	National homicide database (2007–2009), police, judicial, forensic, psychiatric, hospitalization records, toxicological examination. Retrospective study.	-	MMH more often comorbid substance use, mental illness, less likely to commit suicide.	A total of 19.6% hIPF perpetrators committed suicide. Psychiatric care history 41.3%.
[22] (Sweden)	46 hIPF, 165 other homicides by male perpetrators	National homicide database (2007–2009), police, judicial, forensic, psychiatric, hospitalization records, toxicological examination. Retrospective study.	Perpetrator average age 42.5 years (41.4% immigrants), 45.7% employed, 52.2% criminal history. Victim average age 45.5 years (26.1% immigrants), 40.5% employed, 23.9% criminal history.	hIPF perpetrators: older, employed, fewer convictions, stable housing. In total, 80.4% homicides at home, 44.4% with a knife.	19.6% of hIPF perpetrators committed suicide. Post-homicide psychiatric: 8% major mental disorder, 17% personality disorder.
[23] (Sweden)	45 IPF, 9 female perpetrators, 36 male perpetrators	National homicide database (2007–2009), police, judicial, forensic, psychiatric, hospitalization records, toxicological examination. Retrospective study.	Female IPF perpetrators average age 41 years (88.9% unemployed), 55.6% substance use. Male IPF perpetrators average age 43.5 years (28.6% unemployed), 13.9% substance use.	Female IPF perpetrators more likely to have threatened/attacked victim. Male perpetrators: separation/jealousy triggers.	Female IPF perpetrators: higher substance use/personality disorders. Only male perpetrators committed suicide.
[10] (Portugal)	50 hIPF, 27 attempts, 168 IPV	Judicial, forensic records, interviews with perpetrators, questionnaires (BSI, PCL-R, MVI). Analysis of differences between the three groups and multivariate analysis of factors predicting the three forms of violence.	Perpetrators average age 48 years, white, Portuguese, 60% married.	hIPF perpetrators higher socio-economic status, more recent separation, use of a weapon.	No differences in psychopathology, psychopathy. Psychological distress common.
[5] (Portugal)	63 IPF, 12 female perpetrator, male victim	Cases prosecuted for crimes committed 2010–2015. Detailed information from each judicial file on the crime, perpetrator, and relationship. Descriptive analysis of IPF sample and comparison of the two groups.	Male IPF perpetrators: 81% Portuguese, 83% white, 39% employed, 90% below secondary education.	Female perpetrators previously recorded as victims. Male perpetrators more likely to commit suicide after the crime and receive severe sentences. Male IPF perpetrators: 32% criminal history, 62% previous violence, 51% previous separations, motives: 27% suspicion of infidelity, 29% threat of separation, 39% murder with a weapon, 51% premeditated, 15% hid the body.	Substance use history 38% (alcohol 22%), homicide under substance influence 39% (alcohol 2%). History of mental disorder 19% (67% depression, 22% psychosis, 11% dementia). Suicide or attempt 38%.
[24] (Canada)	93 immigrant perpetrators who killed their partners	Death records, risk factors, data from social services. Retrospective study of homicides from 2002–2016 by immigrant perpetrators. Comparison of risk factors in relation to pre-migration trauma, migration stress, and years of citizenship.	50% married, 67% with children at home.	More recent immigrants have fewer risk factors. Pre-migration trauma increases femicide risk.	Pre-migration trauma linked to mental health problems.
[25] (Canada)	135 hIPF	Data from the Canadian Domestic Violence Death Review Committee, which includes information from various sources. Diagnosis of depression either by a specialist or reported by relatives. Comparison of the two groups in the sample.	87 depressed, 48 non-depressed.	64% of perpetrators had depression. Depressed had more risk factors for violence, higher suicide rates.	48.8% committed suicide. Depressed perpetrators had more mental health issues.
[26] (Turkey)	1284 hIPF	Database on violence against women from media publications. Femicides from 2014–2020 and comparison of the period before and after the COVID-19 pandemic.	Majority of femicides by current partners.	57% reduction in hIPF during COVID-19 social distancing, 83.8% during curfews.	-
[27] (UK)	25 hIPF	Detailed information from various sources and relatives. Qualitative study of the stages of the intimate relationship.	-	History of violence, possessiveness, jealousy, separation as trigger.	-
[28] (Denmark)	65 hIPF	Retrospective study of detailed police records, autopsy reports, psychiatric assessments, and toxicological examinations.	Victims, perpetrators often immigrants from Middle East.	69% of perpetrators had no recorded history of violence. Separation, jealousy, infidelity common triggers.	42% had substance use history, 34% under substance influence. Psychiatric diagnosis in 60%.
[29] (USA)	2613 hIPF victims	Data from 17 states, NVDRS records, forensic reports, police reports, toxicological tests, relationship information. Descriptive analysis and variable differences related to crime location.	Victims average age 40 years, mostly white, killed by current partner.	Multiple injuries in 21.6%, firearm most common weapon.	-
[30] (USA)	1693 hIPF	Data from 16 states (2010–2014), NVDRS, and legislation on firearm restraining orders.	-	hIPF incidence 56% lower in states with more firearm legislation.	33% of perpetrators committed or attempted suicide.
[31] (Spain)	168 hIPF	Judicial records (2000–2011). Comparison of immigrant and native Spanish groups.	Perpetrator avg age 42 years, victim younger (37 years), 118 Spanish, 50 immigrants, 47.8% employed.	Jealousy, separation common triggers. Murder often planned; knife frequently used.	Psychopathology history in 73.2%. Suicide attempts in 5.4%.
[32] (USA)	4861 hIPF, 1270 female perpetrator, male victim	Data from NVDRS (2003–2015), 27 states.	Victim usually current partner, white, killed with a firearm.	Jealousy often a motive. Victim under alcohol influence in 20.7%.	No differences between groups in mental illness. Suicide attempt in 46.5% of femicide perpetrators.
[33] (Spain)	307 hIPF	Data from judicial decisions, criminological and psychopathological history of the perpetrator, and relationship information. Comparison of 146 men who killed women after separation (within 1 year) and 161 men who killed their partners without separation.	-	Jealousy, quarrels, recent stressors common triggers.	Substance use in 23.8%, psychosis 8.1%, depression 4.9%.
[34] (Spain)	30 hIPF perpetrators, 20 other femicides	Judicial records, risk assessment questionnaires (RisCanvi). Comparison of femicide perpetrators with other male killers.	Perpetrators average age 42.4 years, 60% Spanish. Victims average age 40.5 years, 68% Spanish.	Intimate partner perpetrators older, less risky behavior, more often killed at home.	Perpetrators under alcohol/substance influence in 33.3%. History of self-harm and suicide attempts in 14.3%.
[35] (Norway)	157 hIPF, 20 female perpetrators	Retrospective study of femicides (1990–2012), judicial, police records, substance use data (and toxicological). Risk assessment questionnaires (DA-R20, SARA, SIVPAS), psychiatric/psychological evaluation. Investigation of risk factors in the two groups.	Perpetrator employed in 47.1%, relationship avg duration 10.86 years.	Jealousy quarrels common motives. Previous violence in 91.1%.	No significant differences in mental health and substance use between male and female perpetrators. In total, 27.4% committed suicide.
[36] (Israel)	194 hIPF	Judicial records of homicides (1990–2010). Retrospective study comparing three ethnic groups.	-	Separation most frequent reason, sexual jealousy common for Ethiopians.	Suicide or suicide attempt more frequent among Ethiopian immigrants.
[37] (USA)	100 hIPF	Data from 100 counties in North Carolina (2002–2011). Official records specifically on domestic violence.	79 urban victims, 21 rural victims.	Higher frequency of hIPF in rural areas, related to economic hardship, fewer services.	-
[12] (UK)	162 hIPF	Data from 2011–2013 England and Wales from police, judicial records, Home Office homicide database, medical, forensic, toxicological records.	72% of perpetrators white, mostly unemployed, cohabiting.	History of previous violence towards victim in 50%.	Suicidal ideation or attempts prior in 40.3%. Alcohol abuse history in 30.7%.
[38] (Turkey)	80 hIPF, 81 other femicides	Retrospective study of femicides in 12 Turkish cities (2000–2010). Data from police, judicial, and forensic records. Comparison of the two groups	Perpetrators average age 39.6 years, 53.2% employed. Victims average age 34 years, 31.2% housewives.	Jealousy, infidelity, honor killing common motives.	10.1% of perpetrators had a history of mental illness.
[39] (UK)	173 hIPF perpetrators	Police records with testimonies, interviews with perpetrators, forensic/toxicological data, judicial documents	Perpetrators average age 41 years, 25% Black, 53.2% employed.	Previous conviction history in 46.2% of perpetrators.	Mental health problems in 19.7% of perpetrators.
[40] (Portugal)	35 hIPF perpetrators, 137 IPV	Retrospective study. Police, judicial, medical records, interviews with perpetrators, risk assessment questionnaires (SARA). Comparison of groups.	Femicide perpetrators average age 48 years, mostly white, low socio-economic status.	Femicide perpetrators had no criminal history in 60%.	No differences in mental illness or personality disorders.
[41] (Sweden)	261 hIPF, 2610 control group	Judicial, criminological, medical records, psychiatric diagnoses based on the National Hospital Register. Comparison of the two groups.	Perpetrators with low educational level 48.7%, immigrants 42.9%.	Femicide perpetrators more often immigrants, lower educational level.	Risk factor for femicide: serious mental illness in 12.6%. 30.7% committed suicide.
[42] (USA)	84 hIPF, 100 other femicides	Data from Wisconsin violent deaths and domestic violence records and indices of residential instability and economic hardship.	Victims married 44.7%. Up to secondary education in 59.5%.	hIPF victims more often married, rural areas, residential instability.	Victims under alcohol influence in 21.2%.
[43] (Ghana)	35 hIPF with subsequent suicides	Study of homicide records from newspapers and interviews with 3 experts.	Perpetrator older by 7 years, married to the victim in 82.9%.	Murder mostly at home, with a firearm.	No indication of mental disorder.
[44] (Finland)	192 hIPF perpetrators, 530 male perpetrator and victim, 44 male perpetrator and male relative victim	Finnish Homicide Monitoring System. Comparison of femicide perpetrators with other groups.	Femicide perpetrators lower socio-economic status, immigrants 5%.	Lower socio-economic status, higher previous criminality.	77% of perpetrators under alcohol or drugs influence.
[45] (Finland)	106 hIPF perpetrators, 445 other female killers, 39 men killed by women, 52 other female killers	Data from multiple sources, police, military, judicial records, psychological evaluations, forensic psychiatric evaluation with hospitalization, PCL-R psychopathy questionnaire. Comparison of groups.	Perpetrators average age 38.3 years, unemployed 53.8%.	Male hIPF perpetrators compared to other male homicide perpetrators were older, more often employed, lower psychopathy, less previous criminality, and substance use.	Personality disorders 71.1%. Psychosis 7.6%. Mood disorder 1%. Alcohol use 62.3%. Intoxicated 81.6%. Substance use 23.6%. Psychiatric hospitalization 32.3%. Self-destructive behavior/suicide attempts 28.2%. 8.5% deemed not responsible.
[46] (USA)	239 hIPF victims	Femicides from 1993–2007. Published records and information from newspapers from government and non-government organizations in Boston, forensic records.	Victim average age 36 years, 72% white, 17% black.	Black women and Hispanic women compared to white non-Hispanic women have a higher risk of being victims of their intimate partner.	-
[47] (UK)	104 hIPF	Qualitative study. Detailed information from various sources and psychiatric history and interviews with perpetrators and relatives of victims.	-	Problems in relationships with women in 75%. Relationship characterized by abuse, controlling behavior, conflicts, jealousy, possessiveness.	Distorted views on relationships, dominant male, submissive female.
[48] (UK)	106 hIPF, 122 IPV	Retrospective study comparing two groups of perpetrators.	White men 85%, African American 5%, Indian and other Asians (3.8–12%).	Femicide perpetrators compared to abusers come from more conventional families, fewer family problems, previous violence towards partner 59%.	Femicide perpetrators less often committed the crime under the influence of alcohol.
[49] (UK)	106 hIPF, 424 male killers	Retrospective study comparing two groups of perpetrators.	-	hIPF perpetrators have a history of school problems, alcohol use, parental divorce, father’s violence towards mother, childhood abuse, previous relationship violence.	Mental health problems in 27.5% of the sample.
[50] (Israel)	15 hIPF perpetrators	Judicial records, in-depth interviews. Qualitative study.	-	Three types of perpetrators: (1) Betrayed, loss of family framework and cultural values, relatively normal. (2) Abandoned, difficulty handling separation, borderline personality traits. (3) Controlling/dominant, loss of chronic dominance/power over the victim, antisocial and narcissistic personality traits.	-
[51] (USA)	57 hIPF, 497 IPV, 208 non-abusive	Interviews with women from different groups and with relatives of hIPF victims, socio-demographic and relationship data. Multivariate analysis of the relationship between pregnancy and abuse.	-	Women during pregnancy and one year after have a lower risk of abuse or femicide.	-
[52] (UK)	90 hIPF perpetrators in prison	Prison records with police, judicial, and psychological assessments. Qualitative study. Content analysis of records. Investigation of 20 variables related to criminality and psychopathology.	55.1% unemployed, 85.6% white British.	36% separation from the victim. Alcohol abuse history and during the murder in 50% of the sample.	15.3% Low Criminality and Low Psychopathology, 49% High Criminality and Low-Moderate Psychopathology, 36.1% Moderate-High Criminality and High Psychopathology.
[53] (Spain)	27 hIPF perpetrators, 131 abusers	Prison and judicial records, questionnaires with a psychologist regarding their beliefs about violence, risk assessment, general psychopathology (SCL-90), Psychopathy Scale, assessment of anger and impulsivity. Descriptive analysis and comparison of the two groups for variables.	Men who killed their partners average age 40 years, divorced or separated (48.1%).	Femicide perpetrators compared to abusers differed only in age, being older. In total, 81.5% had no criminal history.	40.7% had a psychiatric history, mostly related to substance use. Excluded perpetrators with serious mental illness.
[54] (USA)	1322 hIPF	Official homicide records, Chicago Homicide Dataset.	69.2% of victims black, 21.7% white, 7.4% Hispanic.	Risk of femicide higher when man is over 16 years older than victim or woman is at least 10 years older.	-
[55] (Sweden)	164 hIPF, 690 other homicides	Police records, forensic autopsies, forensic psychiatric evaluations in 79% of the sample, PCL psychopathy questionnaire. Retrospective study. Comparison of the two groups.	40% of perpetrators and 30% of victims were immigrants.	hIPF group compared to other killers had higher suicidality, lower psychopathy, less often had other crimes history.	High psychopathology in the sample. 36% psychosis, 11% depression, 38% personality disorders, 5% substance use history.
[56] (USA)	30 women who survived femicide attempts	Interview with victims 5 months to 2 years after the attempt and risk assessment questionnaire. Qualitative analysis.	Victims average age 35 years. 43% African American, 47% white, 7% Latina.	67% history of previous violence from the partner.	-
[57] (USA)	220 hIPF, 343 abused women	Retrospective study in 11 US cities. Police, medical records. Interviews with relatives of victims and perpetrators. Risk assessment questionnaire (DAS). Analysis of risk factors for femicide.	Perpetrator age 34.2 years, 48.9% African American, 22.4% white, 26.5% Latino.	hIPF perpetrators compared to abusers are older, more often African American, unemployed, less educated, with history of alcohol and substance use, gun possession.	Previous threats or attempts of suicide in 25% of perpetrators. Alcohol abuse history in 52% of perpetrators (victims 19.1%), other substances 65.4% (victims 25.3%).
[11] (USA)	380 hIPF and attempts, 384 abused women, 376 non-abused women	Study in 10 cities. Police records and mainly telephone interviews with victims, perpetrators, and relatives of victims. AUDIT questionnaire for alcohol abuse and information on intoxication at the time of the murder. Comparison between groups and the relationship of alcohol and violence.	48.1% of perpetrators unemployed, 48.8% African American.	Compared to other groups, femicide perpetrators were older, African American, unemployed, and victims were African American and had better employment status than perpetrators.	Alcohol abuse by perpetrator associated with higher risk of femicide or attempted femicide.
[58] (Canada)	90 hIPF perpetrators, 50 IPV perpetrators	Judicial, police records, psychiatric report, interview with perpetrator, personality disorder questionnaire MCMI-II. Descriptive analysis of the total hIPF sample and comparison of 50 femicide perpetrators with 50 abusers for personality disorders.	-	Personality disorders more common among femicide perpetrators include passive-aggressive, dependent, self-defeating, and avoidant.	-
[8] (Canada)	705 hIPF	Death records, 1974–1994, judicial, police records, newspapers.	Victims more often minorities and in cohabiting relationships. Perpetrators unemployed.	Motives for murder in 45% threat of separation, 15% jealousy.	Victims under the influence of alcohol or substances in 34%, perpetrators in 43% of murders. Perpetrator suicide in 31%.
[15] (USA)	293 hIPF, 293 other femicides	Forensic, police reports, North Carolina, toxicological examinations, and interviews with police investigators. Descriptive analysis of IPF and comparison with the other group of femicides.	Perpetrator age 47.8 years, 31% of murders committed by ex-partners, 51.2% white.	History of violence towards the partner documented in 66.9%, physical abuse in 76.5%, threats to victim’s life in 83.4%.	In 25.9% of cases, the perpetrator committed suicide. IPF perpetrator under alcohol influence in 47% (victim in 32.3%) or substances (10.7%).

**Table 2 brainsci-15-00589-t002:** Summary of key factors associated with hIPF: Perpetrator and victim characteristics.

Risk Factor	Perpetrator Findings	Victim Findings
Age	Typically aged 34–52, older than victims and other offenders	Often younger than the perpetrator
Employment	High unemployment, some employed	Sometimes better employed than perpetrators; financial difficulty reported in some cases
Education	Often below secondary level	Most had secondary education or lower
Socioeconomic status	Frequently below average	Often economically disadvantaged
Racial/Ethnic background	Overrepresented among migrants, minorities	Black, Hispanic, Indigenous women overrepresented; migrants frequently victims
Mental health issues	Broad range; personality disorders, mood disorders, psychosis	Depression reported in ~14%; mental health data limited
Psychological traits	Higher neuroticism/psychoticism, lower extraversion; subtypes include antisocial, narcissistic, controlling, distressed	Some traits assessed posthumously through interviews from relatives; low levels of neuroticism, psychoticism, high levels of extraversion
Suicide risk	Suicide or attempt common post-homicide (5–50%)	-
Substance use	Common alcohol and drug use (e.g., 10–62%); often intoxicated during homicide	Alcohol use or intoxication reported in 10–66% of cases
Criminal history	Often had past convictions, though less than other homicide offenders	19.1% had criminal history
Violent behavior history	Many had prior IPV reports or convictions	36–91% had IPV history in the relationship
Controlling/jealous behavior	Jealousy, separation, infidelity frequent triggers; stalking and control often reported	Often targets of controlling, jealous, or possessive behavior
Weapon use	Weapon possession a risk factor; knives, sharp objects, firearms common; “overkill” and strangulation often reported	
Relationship status	Usually current or recent partner; cohabiting	
Separation	Often a trigger; many killings followed threats or acts of separation	
Children in household	Presence of children from perpetrator or another father sometimes a risk factor	
Rural vs. urban setting	Higher hIPF in rural settings in some studies	

## Data Availability

No new data were created or analyzed in this study. Data sharing is not applicable.

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
