# Peer review of "Heterosexual Intimate Partner Femicide: A Narrative Review of Victim and Perpetrator Characteristics"

_brainsci, 2025, doi:10.3390/brainsci15060589_

Round 1

Reviewer 1 Report

Comments and Suggestions for Authors

Overall, this presents a valuable contribution to the literature on domestic abuse even more broader than intimate partner homicide. It was well written and of interest to a wide selection of readers.

Just some suggestions for improvement, none of which would individually preclude a recommendation of publishing.

In the table of findings just before Section 4, some of the descriptions of victim-related findings could be more clearly stated as to whether they relate to the victim or the perpetrator (e.g., jealousy and controlling behaviour, history of violent behaviour, and psychological abuse history). I believe these were about the perpetrator's having those three as increasing the risk of homicide rather than the victim's having those experiences, but was not sure.

The text in the paragraph at line 460 about hIPF not being spontaneous appears to conflict with the depiction in the prior table on Premeditation that says that homicide is most often from impulsive acts. Modifying the language to be consistent between then would be helpful.

The typology presented at line 470 is introduced yet left without further support for exactly how the typology was elicited and the potential importance of such an insight.

Author Response

We would like to thank the Editor for considering this manuscript for publication and the two Reviewers for their positive comments and thoughtful and constructive suggestions, which helped to considerably improve the overall quality of the manuscript and provide a more objective interpretation of our results. We provide a detailed point-by-point response to all queries of the Reviewers and have included yellow highlighting in order to mark applied changes (additions) in the body of the revised manuscript.

â–º Response to Reviewer 1 Comments

Comment 1: Overall, this presents a valuable contribution to the literature on domestic abuse even more broader than intimate partner homicide. It was well written and of interest to a wide selection of readers.

Response: We would like to thank the Reviewer for these positive comments!

Comment 2: In the table of findings just before Section 4, some of the descriptions of victim-related findings could be more clearly stated as to whether they relate to the victim or the perpetrator (e.g., jealousy and controlling behaviour, history of violent behaviour, and psychological abuse history). I believe these were about the perpetrator's having those three as increasing the risk of homicide rather than the victim's having those experiences, but was not sure.

Response: We thank the Reviewer for this helpful observation. We agree that the original phrasing could lead to ambiguity. We have revised Table 2 to more clearly specify whether each characteristic refers to the victim or the perpetrator, in particular for the items mentioned.

Comment 3: The text in the paragraph at line 460 about hIPF not being spontaneous appears to conflict with the depiction in the prior table on Premeditation that says that homicide is most often from impulsive acts. Modifying the language to be consistent between then would be helpful.

Response: We would like to thank the Reviewer for pointing out this inconsistency. We have revised the paragraph to clarify that although most hIPFs are not premeditated in a legal or tactical sense, they typically follow a pattern of escalating abuse and coercion. The revised language reflects that while the act itself may be impulsive, it is rarely truly spontaneous in a relational or behavioral context.

Comment 4: The typology presented at line 470 is introduced yet left without further support for exactly how the typology was elicited and the potential importance of such an insight.

Response: We appreciate this comment and agree that further clarification was needed. We have revised the paragraph to propose a broader conceptual grouping derived from the patterns observed across studies, rather than strict typologies. The revised text also explains why this conceptual framework may be useful in understanding perpetrator heterogeneity and in guiding more targeted prevention and intervention efforts.

Reviewer 2 Report

Comments and Suggestions for Authors

This is a very thorough and comprehensive review of multiple studies of intimate partner femicides.  Two particular strengths deserve mentioning.  First, the studies analyzed come from a variety of countries and cultures which enables cross-country comparative examination of similarities and differences.  Second, the paper makes important recommendations pertaining to risk assessment and to prevention.  The latter discussion demonstrates the critical importance of multiple agencies and disciplines needing to work together to confront this global problem.

Author Response

We would like to thank the Editor for considering this manuscript for publication and the two Reviewers for their positive comments and thoughtful and constructive suggestions, which helped to considerably improve the overall quality of the manuscript and provide a more objective interpretation of our results. We provide a detailed point-by-point response to all queries of the Reviewers and have included yellow highlighting in order to mark applied changes (additions) in the body of the revised manuscript.

â–º Response to Reviewer 2 Comments

Comment 1: This is a very thorough and comprehensive review of multiple studies of intimate partner femicides.  Two particular strengths deserve mentioning.  First, the studies analyzed come from a variety of countries and cultures which enables cross-country comparative examination of similarities and differences.  Second, the paper makes important recommendations pertaining to risk assessment and to prevention.  The latter discussion demonstrates the critical importance of multiple agencies and disciplines needing to work together to confront this global problem.

Response: We sincerely thank the reviewer for their positive and encouraging feedback. We are pleased that the cross-national scope of the included studies and the practical implications for prevention and risk assessment were found to be strengths of this review. We appreciate your recognition of the importance of interdisciplinary collaboration in addressing intimate partner femicide, and we hope this work contributes meaningfully to that ongoing effort.